

# On sensor optimisation for structural health monitoring robust to environmental variations

Tingna Wang, David J. Wagg, Keith Worden, and Robert J. Barthorpe

Dynamics Research Group, Department of Mechanical Engineering, University of Sheffield, Mappin Street, Sheffield S1 3JD, UK.

**Correspondence:** Tingna Wang, E-mail: twang71@sheffield.ac.uk.

**Abstract.** Structural health monitoring (SHM) is often approached from a statistical pattern recognition or machine learning perspective with the aim of inferring the health state of a structure using data derived from a network of sensors placed upon it. In this paper, two SHM sensor placement optimisation (SPO) strategies that offer robustness to environmental effects are developed and evaluated. The two strategies both involve constructing an objective function (OF) based upon an established damage classification technique and an optimisation of sensor locations using a genetic algorithm (GA). The key difference between the two strategies explored here is in whether any sources of benign variation are deemed to be observable or not. The relative performances of both strategies are demonstrated using experimental data gathered from a glider wing tested in an environmental chamber, with the structure tested in different health states across a series of controlled temperatures.

Keywords: sensor placement optimisation; structural health monitoring; environmental and operational effects.

## 1 Introduction

Sensor placement optimisation (SPO) is the technique by which the number and location location of sensors is optimised for a specific objective to reduce the cost of an Structural health monitoring (SHM) system without compromising on the effect of monitoring. For this technique, the objective function (OF) should first be designed to evaluate the effectiveness of the data collected from sensors in a given arrangement. Currently, the commonly-used OFs are mainly focussed on maximising the performance of modal identification, parameter estimation schemes (Barthorpe and Worden, 2020; Papadimitriou, 2004; Huan and Marzouk, 2013). However, to improve the ability of an SHM system to identify the structural state, it is necessary to conduct some research on OFs for SPO that are linked directly to structural health-state identification.

One critical aspect of health-state identification is the approach taken to do the damage identification, and specifically whether it makes use of supervised or unsupervised learning. In Worden and Burrows (2001), the authors adopted the normalised mean-square error between the desired results and estimated results from a neural network as a measure of fitness. The sensor layout with the minimum normalised mean-square error was treated as the optimal result. The paper by Samanta et al. (2003), proposed to use the true and false positive rates in the case of a support vector machine (SVM) model used to select the optimal positions using a genetic algorithm (GA). Eshghi et al. (2019) used the detectability of different health states as a criterion to design a sensor network optimally and a surrogate model was applied to reduce the computational burden. In





addition to these non-Bayesian OFs, an approach that utilises an OF based on minimising Bayes risk has also been proposed (Flynn and Todd, 2010). It can be seen that the OF should be adjusted to meet the requirements of a specific project including the type of approach that should be adopted.

Damage is typically indicated via changes in the material properties and at structural boundaries (Farrar and Worden, 2012); these can be revealed via dynamic response properties of a structure, thus realising quantitative global damage detection. However, there are often *confounding effects*, caused by variations in the environmental and operational conditions, which can mask the changes from actual damage. Therefore, a critical step in a damage detection method is to attempt to identify features which are sensitive to variations of material and geometric properties of the structure, but robust to environmental disturbances.

Currently, there are four commonly-used methods to filter out the environmental effects, including; principal component analysis; factor analysis; transformed Mahalanobis squared-distance (MSD) based on independent variables; and cointegration (Deraemaeker and Worden, 2018; Cross et al., 2012). The first three methods are linear techniques in which a linear subspace of a feature space can be identified to account for the environmental conditions (Deraemaeker and Worden, 2018). The remaining subspace of the feature vectors then makes the major contribution to the damage sensitivity. The cointegration technique can help to find common trends among the behaviour of nonstationary signals (Cross et al., 2012), and remove then. When the cointegration between signals does not hold, it indicates that damage may have occurred in the structure. Time series data need to be recorded for the application of this method. Thus, different methods of eliminating environmental effect can be selected based on the characteristics of the accessible data.

This paper develops two strategies for considering temperature variation in the optimum design of the sensor deployment in an SHM system. This technique aims at maximising the damage detection ability of an SHM system by proposing an objective function using a supervised-learning algorithm, namely an SVM. A genetic algorithm is used to search for the optimal sensor deployment with the proposed objective function.

The structure of this paper is as follows: Section 2 demonstrates the process of constructing temperature-insensitive features objectively for a frequency response function (FRF) dataset. Section 3 describes the establishment of the objective function. Section 4 introduces the experiment providing data for the case study in this paper, with the results for the case study discussed in Sect. 5. Finally, conclusions are presented in Sect. 6.

## 2 Feature derivation

Frequency domain data can be employed to reveal changes of vibration characteristics, such as mass and stiffness, of a structure. To generate a robust feature using frequency domain data, according to the literature (Manson et al., 2003), a frequency range (FR) with a specific resolution can be selected to generate features by using distance calculation techniques to compute the discordance between an observation and an observation set, such as Euclidean squared-distance (ESD) and MSD.

In this paper, a feature bagging method is used to objectively derive effective features within a specific FR for FRFs from each sensor. In general terms, a feature-bagging process is an average of all basic features derived from sampled data sets (Murphy, 2012). Here, it is applied by averaging the distance features calculated from $M$ sample data sets of spectral lines





from a relatively-large FR. After obtaining a committee model with $M$ feature values, a combination of these features for an observation from one sensor can be calculated as,


$$D_A^2 = \frac{1}{M} \sum_{m=1}^{M} D_m^2 \tag{1}$$

where $D_m^2$ is the feature (squared distance) value corresponding to the $m^{th}$ sampled set of spectral lines.

To generate $M$ sample data sets of spectral lines, bootstrap sampling is applied among the selected FR to sample spectral lines with repetition. The sampling size $n$ should be set based on the specific data set. The number of sets $M$ can be set based on the total number of spectral lines in the initial frequency range divided by the sampling size. Further detail including
pseudo-codes for the MSD-based feature derivation heuristic may be found in Bull et al. (2019).

Furthermore, consideration of temperature variation will be included in the process of feature derivation. In Sects. 2.1 and 2.2, two approaches are proposed to generate $M$ features by utilising data measured at different temperatures. In order to focus on the effect of temperature, in this paper, the influence of noise is assumed to be negligible. Therefore, high-averaged data are adopted in both approaches.

**2.1 A normalised approach for labelled measurements**

For the laboratory-based studies, if the temperature of a structure can be controlled and recorded, then both the normal condition data and damage data will be labelled with the corresponding temperature. Thus, features for the normal condition data and damage data at each temperature can be calculated respectively. These labels can be used to normalise the temperature on features. On this basis, a distance calculation approach can be taken to calculate features. The noise effect on the high-averaged
data can be ignored, so it is assumed acceptable to use only the mean value of the high-averaged data set to represent the data set itself. As there is thus no uncertainty associated with the feature values, the feature comparisons are between crisp numbers at equal temperatures and it is sufficient to use the ESD as the comparison metric between feature vectors.

Following the methodology given in Eq. (2), the ESD statistic $D_E$ is calculated as,

$$D_E^2 = \sum_{i=1}^{n} (x_i - \mu_i)^2 \tag{2}$$

where $x_i$ is the amplitude value of the $i^{th}$ spectral line, $\mu_i$ is the averaged amplitude value of a spectral line set corresponding to a frequency from an observation set and $n$ is the dimension of the a sample, i.e. the number of spectral lines in a sample.

Note that the ESD-based features here represent temperature-insensitive features obtained by the averaging approach and represent an idealised baseline for comparisons. This option is possible here because of the precise temperature control and recording in the experiments. In general, high levels of noise or limited measurements for averaging will result in uncertainties
on the features which should be taken into account. If sufficient measurements are available to estimate the covariance, the MSD can be used to quantify the discordancy. In practice, for the monitoring of *in situ* structures, the ambient temperature will be uncontrollable and may not be recorded.





## 2.2 A linear approach for unlabelled measurements

In situations where temperature measurements are not available for feature vectors, the influence of temperature still has to
be removed from comparisons. Linear techniques to filter/project out such environmental effects exist, are simple to apply in
practice and computationally efficient (Deraemaeker and Worden, 2018). Furthermore, feature vectors provided by bootstrap
sampling over a large FR are high-dimensional enough to make possible the existence of a linear subspace that can account
for the confounding effects. Such a linear approach, based on the MSD is explored in this paper; it can naturally eliminate
temperature effects as long as the normal condition data include measurements under an appropriate range of temperature
conditions. The MSD is given by,

$$D_M^2 = (\mathbf{x} - \bar{\boldsymbol{\mu}})^\top \mathbf{S}^{-1} (\mathbf{x} - \bar{\boldsymbol{\mu}}) \tag{3}$$

where $\mathbf{x}$ is a vector referring to an observation, $\bar{\boldsymbol{\mu}}$ is the mean value for a set of observations and $\mathbf{S}$ is the corresponding
covariance matrix; $\top$ indicates transpose. Note that the MSD-based features in this paper are used to represent temperature-
insensitive features obtained following application of the linear projection approach of Deraemaeker and Worden (2018).

## 3   Optimisation objective function

To construct an OF having a direct relationship with the damage detection ability of an SHM system, a relationship between
an OF and a classifier distinguishing healthy-state data and damaged-state data needs to be established. For the purposes of
simplicity and computational efficiency, the research here is limited to a linear classifier. The supervised-learning algorithm,
used here to build a linear classifier is the support vector machine (SVM) technique (Vapnik, 2013). The initial reason for the
selection of this classification algorithm is that an SVM makes no assumption about the prior distribution of data, which is
difficult to know exactly in practice.

In order to illustrate the concepts of sensor placement optimisation in this paper, the detection problem itself is deliberately
simplified by considering large 'damage' cases. Furthermore, averaged features are used in order to minimise the effects of
measurement noise. In this situation, the health states of the structure are strictly separable in the feature space. This is a useful
property as it allows a minimal version of the SVM tailored to separable classes and amenable to linear decision boundaries.
This does not represent a restriction on the sensor optimisation problem, as the cases of smaller damage or curved decision
boundaries are both addressable with appropriately adapted versions of the SVM (Vapnik, 2013).

The inputs for a linear SVM training can be represented by,

$$T = \{(\mathbf{x}_1, y_1), (\mathbf{x}_2, y_2), \ldots, (\mathbf{x}_N, y_N)\} \tag{4a}$$


$$\mathbf{x}_i \in \chi = R^n \;\; i = 1, 2, \ldots, N \tag{4b}$$

$$y_i \in \gamma = \{+1, -1\} \tag{4c}$$





where $N$ is the number of training examples, $\mathbf{x}_i$ is the $i^{th}$ feature vector and $(\mathbf{x}_i, y_i)$ is the $i^{th}$ training sample and $y_i$ is either

1 or $-1$ which indicates the class to which the $\mathbf{x}_i$ belongs.

A separating hyperplane can be described by,

$$\mathbf{w} \cdot \mathbf{x} + b = 0 \tag{5}$$

where $\mathbf{w}$ is the normal vector to a hyperplane. The parameter $b/||\mathbf{w}||$ determines the offset of the hyperplane from the origin along the normal vector $\mathbf{w}$, which is shown in Fig. 1.

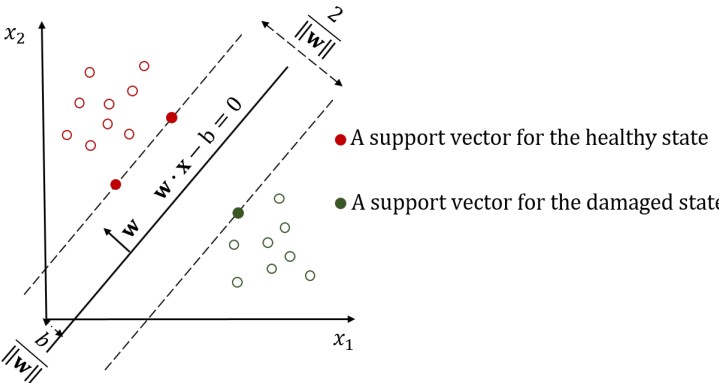

**Figure 1.** The maximum-margin hyperplane for an SVM used as the optimal objective.

Finding the best hyperplane means maximising the smallest signed distance. The smallest signed distance from a sample point $(\mathbf{x}_i, y_i)$ to a hyperplane can be calculated by,

$$\gamma_i = y_i \left( \frac{\mathbf{w}}{||\mathbf{w}||} \cdot \mathbf{x}_i + \frac{b}{||\mathbf{w}||} \right) \tag{6a}$$

$$\gamma = \min_{i=1,\ldots,N} \gamma_i \tag{6b}$$


Maximising $\gamma$ is the optimisation problem,

$$\max_{\mathbf{w},b} \gamma \tag{7a}$$

$$\text{s.t. } y_i \left( \frac{\mathbf{w}}{||\mathbf{w}||} \cdot \mathbf{x}_i + \frac{b}{||\mathbf{w}||} \right) \geq \gamma, \ \ i = 1, 2 \ldots N \tag{7b}$$


This constraint means the signed distance from any feature vector in the training set to the max-margin hyperplane is at least equal to $\gamma$. An important consequence of this constraint is that the max-margin hyperplane is completely determined by those



feature vectors that lie nearest to it. These feature vectors are called *support vectors*. However, changing the values of $\mathbf{w}$ and $b$ proportionally will not change the hyperplane. Therefore, when the ratio of $\mathbf{w}$ and $b$ is constant, $\mathbf{w}$ can be selected according to the demand. For the convenience of calculation, the $||\mathbf{w}||$ can be set to $1/\gamma$. In this case, Eq. (7) can be rewritten as,

$$\max_{\mathbf{w},b} \frac{1}{||\mathbf{w}||} \tag{8a}$$

$$\text{s.t.} \ \ y_i(\mathbf{w} \cdot \mathbf{x}_i + b) - 1 \geq 0, \ \ i = 1, 2 \ldots N \tag{8b}$$

Therefore, as shown in Fig. 1, a maximum margin width is equal to,

$$D_{margin} = \frac{2}{||\mathbf{w}||} \tag{9}$$

For different input data sets, the SVM will provide different best hyperplanes that match different maximum margin widths. Because there are no hyperparameters to be optimised in this process, the OF can be evaluated quickly. It is beneficial to search out the best sensor combination from a large number of sensor candidates.

In this research, the combination of sensor locations which provides feature vectors that make the healthy-state data and the damaged-state data most separated will be selected as the optimum; i.e., the sensor deployment with the largest max-margin obtained by an SVM. To find the optimal deployment of sensors, an integer GA is adopted here, which can search out the globally-optimal result with variables that are integer-valued. Linear constraints are used to make sure that non-repetitive sensors exist in an optimal sensor deployment. These constraints refer to the fact that the difference between any two selected sensor indices should not be less than 1, which can be expressed as follows,

$$k_i - k_{i+1} \geq 1, \ \ i = 1, \ldots, K-1 \tag{10}$$

where $k_i$ is the index of a selected sensor and $K$ is the number of selected sensors.

## 4  Experiment set-up and design

The structure under investigation is a glider wing (shown in Fig. 2). Figure 3 is a schematic showing the 36 candidate sensors used with their labelled positions drawn to scale. These sensors are evenly distributed on the wing structure to provide the candidate sensor-position combinations. To simulate a reversible damage scenario rather than inflict permanent damage on the wing, the damage was introduced by adding mass blocks at discrete points. The first mass block (M1) was added between sensors 4 and 5 and had a mass of 860 g. The second mass block (M2) was added between sensors 6 and 7 with a mass of 900 g. As mentioned earlier, these represent quite large damage in order to produce separated feature clusters. The locations for both mass blocks are shown in Fig. 3. Three damage cases are considered: mass addition at locations [M1], [M2] and [M1, M2].





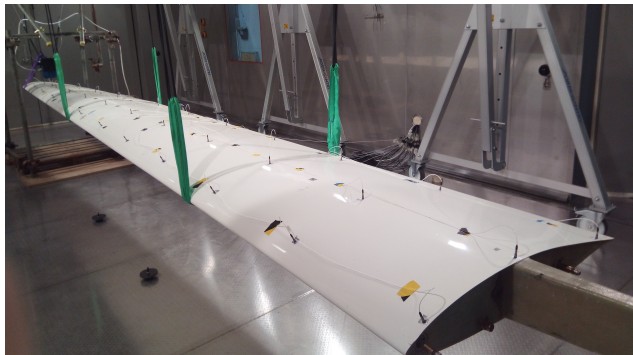

**Figure 2.** Photograph of the experiment setting in the testing chamber.

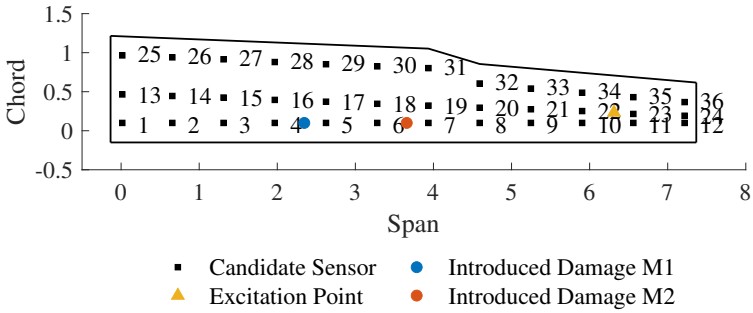

**Figure 3.** Labelled positions of experiment sets on the gilder wing.

The wing was excited at a point between sensor 22 and sensor 23, as shown in Fig. 3, by the ETS solutions VT100 electro-dynamic shaker. A Gaussian white-noise excitation was generated within the Siemens LMS acquisition system and amplified using the ETS solutions LA500 power amplifier. The FRFs were measured using PCB resonant piezoelectric accelerometers

and sampled using a 64-channel acquisition system controlled by LMS software. Each FRF is an averaged value of eight measurements to make the obtained FRFs smoother. The frequency range over which the FRFs were taken is 0-4096 Hz. The frequency resolution is 0.25 Hz.

Seven different temperatures in the chamber were controlled and recorded, ranging from 0 °C to 30 °C at intervals of 5 °C. Two measurements of the FRF matrix were recorded for the structure under the normal condition at each controlled

temperature, with only one measurement for the structure under damaged condition at each temperature. Figures 4 and 5 show the FRFs from one sensor – number 17 – which indicate that the effects of temperature and damage on FRF are almost the same order of magnitude. Therefore, involving the temperature in the SHM system design process is necessary.

The feature-bagging method was conducted to generate features using measurements with or without temperature labels in this case study. The sampling size $n$ should be less than the number of observations for the normal condition to avoid a singular

covariance matrix in the MSD calculation. Because there are two observations for each normal condition state at each controlled

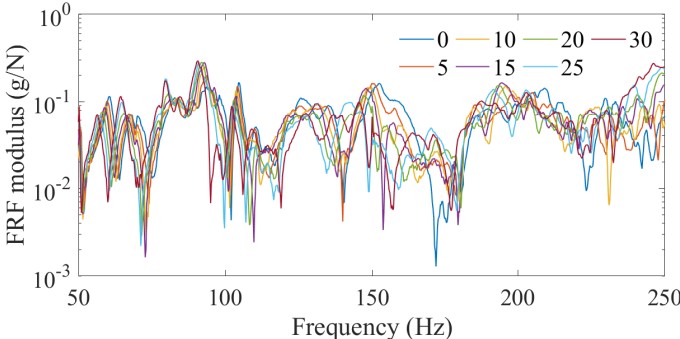

**Figure 4.** An example of temperature effect on frequency response: Normal condition FRFs collected at seven different temperatures from sensor 17.

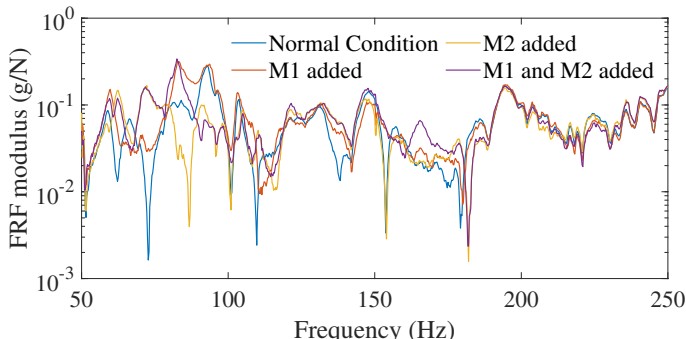

**Figure 5.** An example of the adopted frequency range: FRFs between 50 Hz and 250 Hz collected at 15 °C from sensor 17 for four condition cases.

temperature and data for seven temperatures were recorded, the total number of observations for the normal condition is 14. Therefore, the sampling size $n$ was set to 10 and the covariance determinants were checked to avoid the singular matrix. The number of samples $M$, was set to be equal to or slightly larger than $d/n$, where $d$ is the dimension of the original FR.

By plotting FRFs for the structure in different healthy states, a FR from 50 Hz to 250 Hz is selected to generate the sample
data sets of spectral lines. Here, FRFs for the structure with the temperature at 15 °C are used as an example. As can be seen in Fig. 5, this frequency range is effective because it is sensitive to all damage cases, i.e. it is easy to distinguish the FRFs for these three damage cases from the FRF of the normal condition case. The resolution of the FRFs is 0.25 Hz, so the dimension of the original FR space is 801. Thus, the number of samples can be set to 90 here.

The same sampling size and number of samples are taken for the derivation of the ESD and MSD-based features, which is
beneficial for comparing the optimal results when different features are extracted from the same data set. In addition, only one discordance measure is calculated for one observation from each sensor. Selection of a single feature for each sensor based on a relatively-large frequency range is attractive as it leaves aside the feature selection task.



Because the measured data in this test are high-averaged and the size of introduced damage is large, the normal condition data and damaged state data used for training an SVM classifier are linearly separable (as discussed earlier). Therefore, the proposed OF constructed by a linear SVM for a linearly separable case can be applied here. The misclassification rate is also calculated to check the separability. It is obvious that if there is a sensor combination providing a non-zero misclassification rate, it should be abandoned directly. Although three damage cases are considered with mass additions at locations [M1], [M2] and [M1, M2], only binary classification is considered here to distinguish the normal condition case and the three damage cases collectively. In this way, the relationship between the damage position and the sensor deployment can be studied.

## 5  Results and discussion

### 5.1  Feature bagging results and analysis

In order to improve the efficiency of the calculation, it is necessary to normalise the feature vectors before training a classifier by mapping the min and max values of all given dimensions to 0 and 1. The feature-bagging results of ESD-based features and MSD-based features for the normal condition case and three damage cases from 36 sensors at seven different temperatures are shown in Figs. 6 to 9.

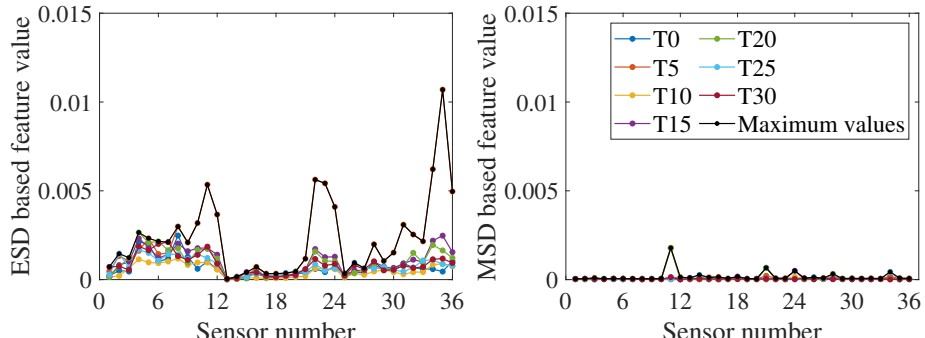

**Figure 6.** Feature-bagging results based on ESD (left) and MSD (right) for the normal condition. Seven controlled temperatures T0 – T30 and the maximum value among the features for seven different temperatures are shown by colour markers. ESD-based features refer to features obtained by the normalised approach with temperature labels. MSD-based features refer to features obtained by the linear approach without temperature labels.

By analysing the positions of sensors providing comparatively large discordance values in different damage cases, it is apparent that data collected from sensors close to the added mass blocks are sensitive to this damage as may have been expected. For example, two of the largest discordance values calculated via the ESD or MSD for damage cases 1 come from sensors 4 and 5, adjacent to the mass block M1. Furthermore, by comparison of Fig. 6 and Figs. 7 to 9, it can be seen that the influence of temperature on the two types of features extracted from a subset of sensors is much smaller than the influence of





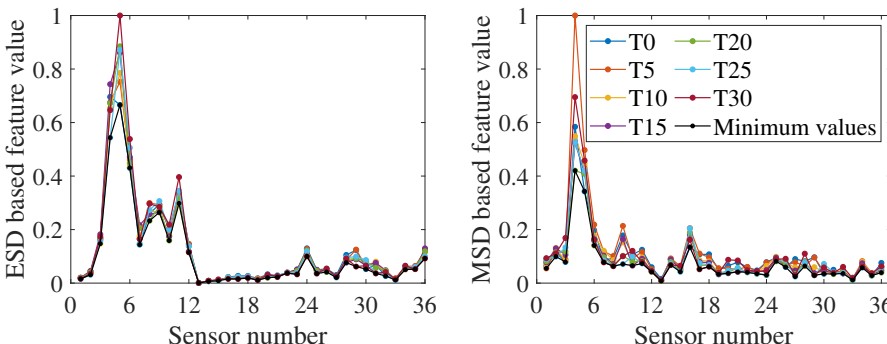

**Figure 7.** Feature-bagging results based on ESD (left) and MSD (right) for the damage case with M1 added. Seven controlled temperatures T0 – T30 and the minimum value among the features for seven different temperatures are shown by colour markers.

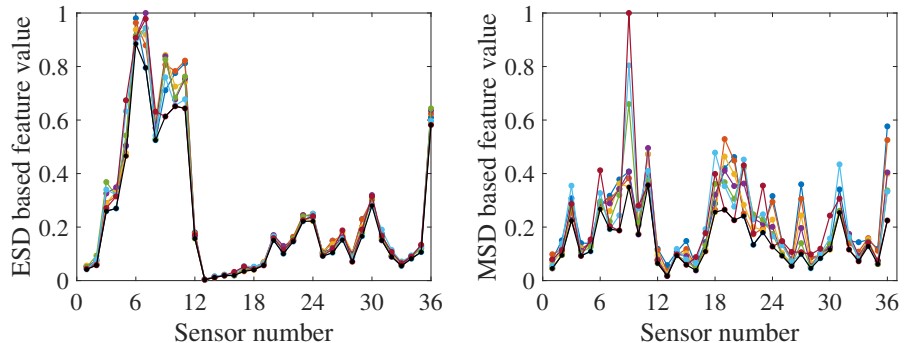

**Figure 8.** Feature-bagging results based on ESD (left) and MSD (right) for the damage case with M2 added.

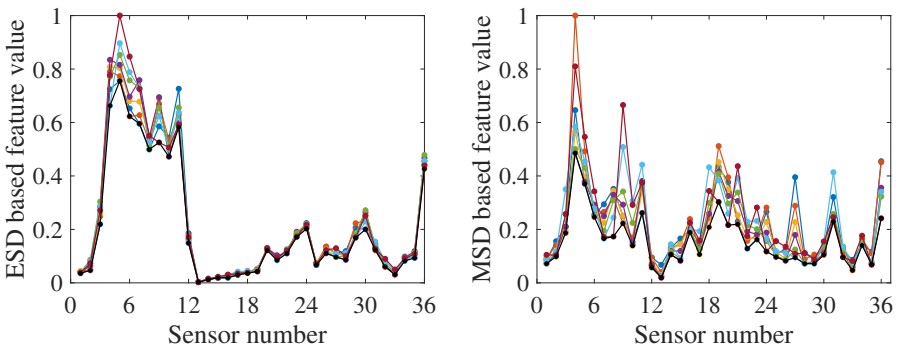

**Figure 9.** Feature-bagging results based on ESD (left) and MSD (right) for the damage case with M1 and M2 added.





the damage. This observation suggests that both feature extraction techniques can help to provide effective features robust to temperature variations but sensitive to the damage.

Furthermore, from Figs. 7 to 9, it can be seen that a sensor at a fixed location on the gilder wing has different sensitivity to the occurrence of the same damage at different temperatures. A reasonable explanation is that at different temperatures, the
physical parameters of various structural components (such as stiffness and cross-sectional area) will change. This variation results in the same part of the structure responding differently to the same damage at different temperatures. Thus, the effect of one type of damage on different locations of the structure at a certain temperature cannot be applied after the temperature of the structure changes. In addition, because the minimum feature values are consisted of features for different temperatures, it is impossible to set one temperature as the most unfavorable temperature for damage detection. Therefore, it is necessary to
involve the temperature when analysing the structural response and training a classifier.

## 5.2 Optimal results based on SVM models

The GA is used to optimise sensor sets containing between one and five sensors for detecting three different damage cases separately by using ESD or MSD-based features. The resulting optimal combinations are listed in Tables 1-3. It can be seen that sensor locations selected by the SVM optimisation technique using ESD-based features are mainly distributed on the
leading edge of the glider wing, while the results for the MSD-based features show that optimal sensors are much more scattered on the structure.

**Table 1.** Optimal sensor combination designed for M1 case detection and corresponding max-margin widths.

| Num.of sensors | ESD | Max-margin width | MSD | Max-margin width. |
|---|---|---|---|---|
| 1 | 5 | 0.6634 | 4 | 0.4202 |
| 2 | 4, 5 | 0.9597 | 4, 5 | 0.5859 |
| 3 | 4,5,6 | 1.0511 | 4,5,16 | 0.6156 |
| 4 | 4,5,6,11 | 1.1030 | 4,5,16,25 | 0.6210 |
| 5 | 4,5,6,9,11 | 1.1411 | 4,5,6,16,25 | 0.6367 |

**Table 2.** Optimal sensor combination designed for M2 case detection and corresponding max-margin widths.

| Num.of sensors | ESD | Max-margin width | MSD | Max-margin width |
|---|---|---|---|---|
| 1 | 6 | 0.8819 | 11 | 0.3549 |
| 2 | 6, 7 | 1.2582 | 9, 19 | 0.5775 |
| 3 | 6,7,11 | 1.4548 | 9,11,20 | 0.7082 |
| 4 | 6,7,9,10 | 1.6012 | 9,11,19,36 | 0.8467 |
| 5 | 6,7,9,10,11 | 1.7229 | 9,11,19,20,36 | 0.9283 |



**Table 3.** Optimal sensor combination designed for M1M2 case detection and corresponding max-margin widths.

| Num.of sensors | ESD | Max-margin width | MSD | Max-margin width |
| --- | --- | --- | --- | --- |
| 1 | 5 | 0.7534 | 4 | 0.4859 |
| 2 | 4, 5 | 1.0436 | 4, 19 | 0.6395 |
| 3 | 4,5,11 | 1.2666 | 4,11,19 | 0.7438 |
| 4 | 4,5,6,11 | 1.4115 | 4,5,11,19 | 0.8344 |
| 5 | 4,5,6,7,11 | 1.5431 | 4,5,11,19,36 | 0.9070 |

By comparing the selected sensor locations in Tables 1 to 3 with the minimum values of features shown in Figs. 7 to 9, an expected phenomenon is observed: generally, locations corresponding to the larger minimum feature values are selected as the optimal sensor locations. However, it can also be seen that the distribution of feature vectors from one sensor for different temperatures can also affect the results of the SVM optimisation technique.

To Demonstrate this visually, data from two sensors (9 and 19) selected by the SVM optimisation technique with MSD-based features adopted to detect damage case 2 in Table 2 and two sensors (9 and 11) providing the two largest minimum-feature values for the same damage case in Fig. 8 (left) are used to build two classifiers and calculate the maximum margin widths. The results are shown in Figs. 10 and 11. The max-margin width calculated by the data from sensors 9 and 19 is 0.58, which is larger than the margin width (0.56) corresponding to sensors 9 and 11.

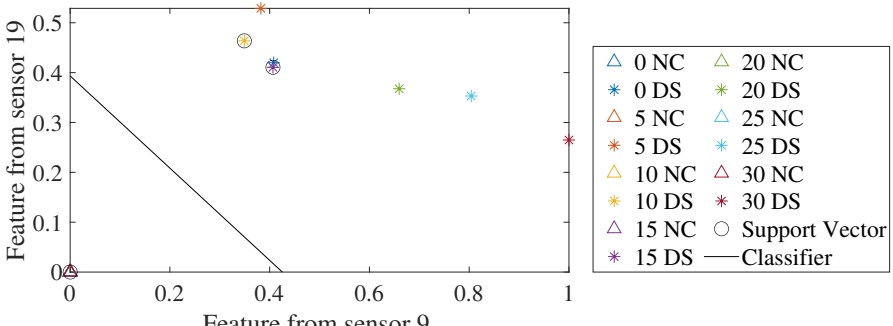

**Figure 10.** Distribution of MSD based features from two optimal sensors selected by the SVM optimisation technique. $\triangle$ refers to the normal condition and $\ast$ refers to the damaged state.

To explore the relationship between the max-margin width corresponding to the optimal sensor combination and the number of selected sensors, the results for the M1 case are taken as an example and demonstrated in Fig. 12. Here the relative position of the line for SPO with temperature labels is higher than those without temperature labels. A similar phenomenon also occurs for the M2 case and M1M2 cases; this indicates that the linear method without using the temperature labels employed to eliminate the influence of temperature will sacrifice sensitivity to damage to a certain extent. This observation is also in line with the expectation that more information can provide better results.



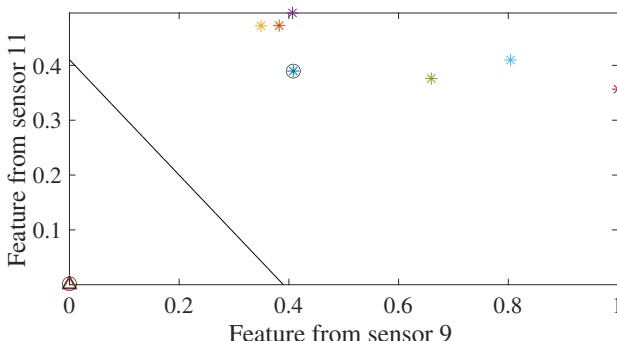

**Figure 11.** Distribution of MSD based features from two sensors providing the two largest minimum - feature values.

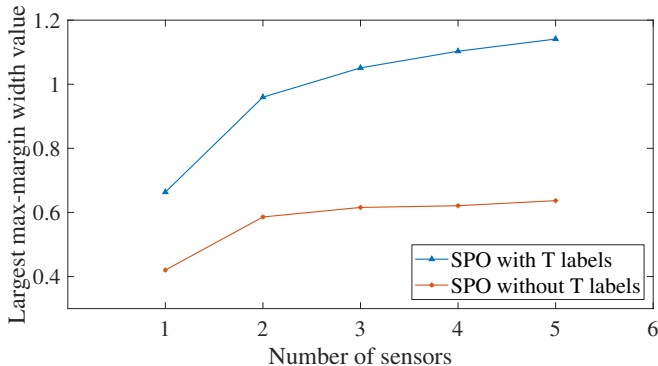

**Figure 12.** Optimal max-margin widths and corresponding number of selected sensors for M1 case detection

However, one obvious advantage of the linear method is that it is not necessary to strictly control the temperature of a structure to a specific degree, which can be a tricky and/or expensive process even in the limited number of cases where it is feasible. Additionally, the number of measurements can be greatly reduced if the normal condition data for all temperatures is used to provide a non-singular covariance used in distance calculation. This condition helps to reduce the cost and time of data collection. Therefore, there is a trade-off between the cost and feasibility associated with temperature control and label collection, and the lack of sensitivity associated with the linear method to address the confounding effects conveniently.

A method that may warrant investigation, is to partition a large temperature range into several segments, with the linear approach applied within each segment and the normalised approach used to combine the feature values for each temperature segment with the SVM optimisation technique to process SPO.





# 6 Conclusions

This paper illustrates two SPO techniques designed for damage detection while taking into account temperature effects; the key contributions are: (1) to investigate two approaches for extracting damage-sensitive features in the presence of either recorded or unrecorded temperature variations and, (2) to investigate appropriate optimisation functions for evaluating the resulting
sensor combinations.

A case study of a glider wing shows that, compared to the normalised method using the temperature label, the linear method that did not require temperature labels provided features that were less sensitive to damage. However, it is cheaper and more convenient to extract temperature-robust features in practical engineering. Meanwhile, the proposed optimisation criterion – maximum margin width – is an effective criterion considering the damage detection ability of the designed SHM system,
provided that the health-state classes are separable in the feature space. If this condition is not satisfied, the criterion should be extended to soft-margin SVM classifiers.

Further work has been considered. A further experiment will be conducted on this glider wing to collect more measurements in one case. A smaller size of the introduced damage will be selected to provide a more challenging dataset (i.e. not separable). Then test data-set can be obtained to provide an unbiased evaluation of a damage detection system fit on the training data-set.

*Data availability.* The data that support the findings of this study are available from the corresponding author, TW, upon reasonable request.

*Author contributions.* This research was conceptualized by TW and RB. Funding acquisition was achieved by DW, RB and KW. The data management, methodology, formal analysis and original draft preparation were performed by TW, who was supervised by DW, RB and KW. The manuscript was reviewed and edited by all authors.

*Competing interests.* The authors declare no conflicts of interest.

*Acknowledgements.* The authors would like to acknowledge the support of Siemens Gamesa and UK EPSRC via grant number EP/R004900/1. This research made use of The Laboratory for Verification and Validation (LVV) which was funded by the EPSRC (grant numbers EP/J013714/1 and EP/N010884/1) and the University of Sheffield.



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
