# Peer review of "On sensor optimisation for structural health monitoring robust to environmental variations"

_Wind Energy Science, 2021_

## Author Comment (AC1)

**Reply on RC1**

In this paper, the authors have proposed two SHM strategies for considering temperature variation in the sensor placement optimisation problem. The proposed technique employs a genetic algorithm by introducing an objective function based on a linear support vector machine classifier in order to maximise the damage detection ability of an SHM system. The model is well-structured, and the results sound promising. The writing quality of the paper is also nice. The reviewer has some comments which would like the authors to consider in the revision of the paper:

| | |
|---|---|
| • The authors have claimed that their proposed SHM method is robust to environmental variations but only have considered temperature variation. The robustness of the method to the other parameters, such as wind speed, should be discussed, and if necessary, be evaluated by considering these parameters. | The effect of other parameters can also be considered in a similar way as to the temperature. Depending on the availability of the parameter labels, one of the two proposed techniques, or a combination of the two, can be adopted.

The focus of the paper is on experimental data from a real structure, and the tests are quite demanding. The current tests on a glider wing have only considered the temperature factor, in order to propose a general framework considering SPO robust under environmental variations. The LVV facility would allow tests at different wind speeds, so tests could be carried out, but they would require to be scheduled and resourced, and this is unfortunately not possible in the immediate future.

Some discussion has been added to the paper including in lines 42, 45 -47. |
| • The proposed method selects different sensors in different damage cases (based on the damage location). In other words, there is no unique solution for all states. Therefore, in order to reach an optimum selection, the data from all sensors should be | In this paper, a support vector machine (SVM) for binary classification is introduced as a simple example to show the optimisation criterion and realise the classification tasks.  If more than two states need to be considered, a multi-class SVM or other multi-classification |

used in each damage case. What can be done to reduce the number of sensors (not dependent on the damage location) in real applications where the damage source is unknown? Please explain about this issue.

algorithms can be used; this does not affect the proposed framework. An appropriate criterion can be adopted as the optimisation objective according to project requirements.

If only data sets without labels for different states are available, an unsupervised algorithm and corresponding criterion can be taken to replace the classification algorithm and the criterion used in the proposed SPO framework.

If only the data for the healthy state are available, the possible solution is to make the data from the selected sensors contain as much useful information as possible. The definition of 'useful information' can be determined by application requirements. Another possibility is to model the structure of interest and validate the model on healthy-state data. Based on this, sensors can be optimised to better calibrate the model. These are substantial developments that form part of our planned future work.

In real applications, the SHM process would have an 'operational evaluation' stage, in which the likely sites and types of damage would be assessed: e.g., by looking for 'hot spots' like stress concentration. Once the potential locations of the damage are identified, one can try and generate data by modelling or by using proxies.

The changes in the paper include those in lines 201-205, 271-272.

| | |
|---|---|
| • There are some typos within the text that the authors need to correct in the revised version of the paper:
• In line 11, "number and location **location** of sensors"
• In line 81, "n is the dimension of **the** a sample"
• In line 231, "To **D**emonstrate" | These typos have been corrected. |

---

## Author Comment (AC2)

**Reply on RC2**

The paper deals with the optimisation of sensors for damage identification considering the temperature which is one of the main parameters that induce changes in structures. A method is proposed based on an objective function which is defined by using a support vector machine (SVM). Additionally, the presented experiment shows the importance of understanding the environmental conditions in structural health monitoring for reliable damage assessment.

The paper is well written and organised. The quality of the experiment is high, and the results are clearly presented, and the graphics are readable. The minor corrections are already made with the suggestions of the first reviewer. I would not suggest any other corrections.

| | |
|---|---|
| The reviewer would like to ask if the proposed method is also applicable in the SHM systems where the structures are monitored continuously in the existence of temperature changes. | Yes, it can. The aim of this study is to establish a general framework to consider a certain environmental effect in the design process of a sensor system. After the sensor system is put into use, the healthy state data should be collected at different temperatures for a period of time firstly. Then, the continuously collected data can be used for robust feature extraction using the methods mentioned in the article. The time interval for extracting features should be set according to the regulations of the project. In addition, the algorithm used for damage identification can be adjusted according to specific requirements. |
| In other words, by using the proposed feature extraction methods, is it possible to detect sensitivities to the damage by using a set of measurements that have frequency variations caused by temperature? | Yes, it is. A set of healthy state measurements with frequency variations caused by temperatures can be used as a baseline to extract features. In fact, it is possible to remove temperature (or other benign) effects, even without temperature measurements. There is a very large body of work on this topic – data normalisation. However, this paper looks at a more fundamental problem; if the sensor network delivers robust data, there is much less work upstream in removing temperature effects etc. |